

# Vegetation dynamics of abandoned paddy fields and surrounding wetlands in the lower Tumen River Basin, Northeast China

Guanglan Cao[1], Kazuaki Tsuchiya[1], Weihong Zhu[2,3] and Toshiya Okuro[1]

[1] Department of Ecosystem Studies, Graduate School of Agricultural and Life Sciences, University of Tokyo, Tokyo, Japan
[2] Department of Geography, Yanbian University, Yanji, Jilin, China
[3] Jilin Provincial Joint Key Laboratory of Changbai Mountain Wetland and Ecology, Changchun, Jilin, China

## ABSTRACT

On the temperate lowland plain of the lower Tumen River, agricultural development has converted most marshland into paddy fields. However, the locations of old paddy fields in the lowland temperate zone, where the vegetation structure is dominated by herbs adapted to seasonally wet or waterlogged conditions, are poorly known, and the impact of land use history on marshland diversity and shifts in plant functional groups has been scantly researched. In this study, we used a chronosequence approach to investigate herbaceous wetland communities in different recovery phases (<5 years, 5–15 years, and >15 years), as well as natural wetland as a reference. We assessed their ecological characteristics, species composition and diversity to determine how they change during natural succession. Plant species composition and dominance in the abandoned fields changed markedly during natural secondary succession. Initially, the annual weeds *Echinochloa crus-galli* and *Bidens tripartita* were dominant. Later, communities gradually became dominated first by *Polygonum thunbergii* and then by tussock-forming *Carex rostrata*. Species diversity was higher in abandoned fields than in natural wetlands and decreased with time. The partition of β-diversity components revealed that replacement was the prominent process structuring plant communities in paddy field at different times since abandonment. Our results suggest that the vegetation of abandoned paddy fields could be restored effectively through natural succession, although there were some differences in plant functional group traits. Abandoned paddy fields may be good sites for restoration of wetland species and conservation of wetland habitat.

# INTRODUCTION

Wetlands are among the world's most productive and valuable ecosystems (*Mitsch & Gosselink, 2015*; *Kennedy & Mayer, 2002*). They are important habitats for a variety of flora and fauna, and are vital ecosystems that provide diverse services. They not only are essential for ecological functions such as maintaining biodiversity, controlling floods,

Corresponding author
Toshiya Okuro,
aokuro@mail.ecc.u-tokyo.ac.jp

and removing pollutants, but also serve necessary economic functions in rice and fish production, transport, and hydropower energy (*Costanza et al., 1997*; *Mitsch & Gosselink, 2015*; *Zedler & Kercher, 2005*). Despite this, they have suffered a loss of 54%–57% of their area worldwide, which continues to be under pressure from agriculture, urban expansion, industrialization, and resource extraction (*Davidson, 2014*; *Zedler & Kercher, 2005*).

China, with 10% of the world's wetland area, has suffered great losses and degradation, largely attributed to agricultural intensification, severe population pressure, and misguided policies (*An et al., 2007*; *Wang et al., 2012b*; *Sun, Liu & Li, 2006*). Effective protection and restoration of damaged and degraded wetlands is becoming more and more urgent.

Abandoned paddy fields provide opportunities to restore wetlands and serve as substitute habitats for wetland species (*Lee, You & Robinson, 2002*; *Kusumoto, Ohkuro & Ide, 2005*; *Takanose et al., 2013*; *Yamanaka et al., 2017*). They may be good sites for restoration of wetlands and conservation of wetland habitats (*Cho, Lee & Lee, 2018*).

Natural restoration of vegetation in abandoned paddy fields is an example of secondary succession (*Cramer & Hobbs, 2007*). Most studies have focused on large-scale farmland, upland fields, and tropical and subtropical areas (*Lee, 2006*). Few studies have focused on secondary succession of old paddy fields in the lowland temperate zone, where the vegetation structure is dominated by herbs (grasses, sedges, and forbs) adapted to seasonally wet or waterlogged conditions. Thus, our understanding of temperate lowland ecological communities is limited.

The lower Tumen River is situated on the lowland plains of temperate north-eastern China, which have vast natural, wetland-dominated, seasonal herbaceous communities, which serve as important migratory routes for water birds (*Brinson & Malvarez, 2002*). Since the 1980s, conversion of wetlands to paddy fields or fishery ponds has eradicated more than half of the area (*Zheng et al., 2017*). Alterations to the hydrological regime and development are the major threats to this ecosystem and can lead to habitat destruction and shifts in community function.

Here, we selected abandoned paddy wetlands dominated by seasonal herbaceous communities at different recovery phases and assessed their species composition, species diversity and plant functional groups to determine how they change over time. The objectives were (1) to determine whether the abandoned paddy fields were self-regenerating; (2) to assess their stage of secondary succession and how long it took to restore vegetation; and (3) to investigate vegetation succession patterns in abandoned paddy fields.

## MATERIALS & METHODS

### Study sites

The study area is located in the lower Tumen River Basin, in north-eastern Jilin Province, China (42°25′20″–43°30′18″N, 129°52′00″–131°18′30″E; 5–15 m a.s.l), and covers the international boundaries between China, North Korea, and Russia (*Zhu et al., 2012*). The area has a mean annual temperature of 5.65 °C, a maximum monthly average of 21.2 °C in August, and a minimum monthly average of −11.7 °C in January. Mean annual precipitation is 606.8 mm, of which approximately 70% falls during June–September (*Kang et al., 2017*).

This area has a diverse array of wetlands totalling 8,054 km$^2$ (*Cui & Yang, 2001*). The Tumen River Basin is characterized by a typical temperate monsoon climate zone and is usually inundated seasonally by rain, and then dries out (*Gao, Zhu & Wang, 2000*).

## Survey design

We used a chronosequence approach to investigate herbaceous wetland communities during secondary succession in recovery phases of <5 years, 5–15 years, and >15 years since the last cultivation. A natural wetland region was selected on the lower Tumen River. Since it was difficult to find an undisturbed natural wetland as a reference site, we chose a less disturbed one that had been uncultivated for more than 40 years. Information on the age of abandoned sites was collected via interviews with land owners and village heads. All sites were flat and subject to similar hydrological conditions (*Guo et al., 2017*).

## Vegetation sampling

To gather comprehensive information on the wetland vegetation, we surveyed vegetation by using the plot method and quadrat method (*Magee et al., 1999*; *Ruto et al., 2012*). We located twenty six 100 m$^2$ plots with different phases of succession, and randomly selected five 1 m$^2$ quadrats within each plot. To avoid spatial autocorrelation between plots, the plots were separated by at least 1,000 m, but remained within the same general landscape position. Each site was surveyed once during August 2016, the peak growing season in the region. We recorded species composition, species density, species coverage, plant height, water depth, latitude and longitude, wetland type, and habitat details at a total of 130 quadrats. We used species richness (the number of species in each quadrat) as a measure of plant diversity and recorded the abundance of individual species within each quadrat. The scientific names of all vascular plant species complied with the Y List based on APG (http://ylist.info/, queried in November 2014; Japanese names) and with a Chinese database (http://www.plant.csdb.cn/; Chinese names).

## Data analysis
### Plant diversity analysis

We calculated four species diversity indices: Margalef's index of species richness R, Shannon–Wiener diversity index H, Simpson's index of diversity D, and Pielou's evenness index J (*Krebs, 1989*). All have low to moderate sensitivity to sample size and are widely used (*Alsterberg et al., 2017*; *Cao & Zhang, 1997*; *Magurran, 1988*).

### Margalef's index of species richness, R

Margalef's R is a simple measure of species richness (*Margalef, 1958*):

R = number of individuals of a species

### Shannon–Wiener diversity index, H

The α-diversity of species within a community or habitat was calculated as the Shannon–Wiener diversity index (*Shannon & Wiener, 1949*):

$$H = -\sum_{i=1}^{s} P_i \log P_i. \tag{1}$$

where $p_i = s/n$, $s = $ number of individuals of a species, $n = $ total number of individuals in the sample, and logarithm is in base $e$.

### Simpson's index of diversity, D
Simpson's D was calculated as:

$$D = 1 - \sum_{i=1}^{s} P_i^2 \tag{2}$$

### Pielou's evenness index, J
Pielou's J indicates the evenness of species (*Pielou, 1966*):

$$J = H/lnS \tag{3}$$

where $H = $ Shannon–Wiener diversity index and $S = $ total number of species in the sample.

### Replacement, richness difference and nestedness indices
The SDR simplex approach was used to estimate the relative importance of $\beta$-diversity components and similarity by partitioning the pairwise gamma diversity into three components (*Podani & Schmera, 2011*). Abundance (i.e., cover) data was used to calculate Ružička similarity (S), abundance replacement (R), and abundance difference (D) by the SDR-abunSimplex programme (*Podani, Ricotta & Schmera, 2013*). The SDR results were graphed with a ternary plot using the Ternary Plot option in the NonHier routine of the SYN-TAX 2000 package (*Podani, 2001*). In the ternary plot, each vertex corresponds to one index (S, D, or R).

## Statistical analyses
For statistical analyses we defined wetland species (W) as those that usually occur in wetland, marshland, ponds, or rice fields, on shorelines, or under water, and non-wetland species (NW) as those without any description of wetland habitat.

Raunkiaer's life-forms were used to categorize plant species as follows (*Raunkiaer, 1934*): hydrophyte (HH), hydrophyte therophyte (HH(Th)), hemicryptophyte (H), therophyte (Th), therophyte (winter annual) (Th(w)), geophyte (G), and chamaephyte (Ch).

Patterns of chorological spectra were quantified at species level and categorized as Cosmopolitan, Holarctic, Palaearctic, Afrotropical, Oriental, or Australian (*Takhtajan, 1986*).The Holarctic region comprises two subregions: Palaearctic (Europe, Asia north of the Himalayas, and northern Africa) and Nearctic (North America excluding southern Florida, and Greenland (*Katinas, Morrone & Crisci, 1999*).

For illustrating wetland patterns and the floristic relationship with sites, ordination methods are more appropriate than clustering when the sites come from different land use forms. Therefore, we assessed variations in plant species composition and distribution patterns among vegetation communities using detrended correspondence analysis (DCA) ordination based on community coverage. To examine the changes in the different functional groups during succession, plant species were grouped according to life form

**Table 1** Summary of the attributes of the sampling plots for the abandoned paddy wetlands in the different restoration years in downstream of Tumen River Basin, Northeast China.

| Items | Years since abandonment | | | Natural wetland | Total |
|---|---|---|---|---|---|
| | Ab < 5 | 5 < Ab < 15 | Ab > 15 | | |
| Site number | 6 | 5 | 4 | 11 | 26 |
| Sample number | 30 | 25 | 20 | 55 | 130 |
| Sample area (m²) | 30 | 25 | 20 | 55 | 130 |
| Family number | 22 | 20 | 15 | 24 | 39 |
| Genera number | 40 | 33 | 25 | 38 | 74 |
| Species number | 62 | 42 | 35 | 47 | 114 |
| Plant coverage (%) | 85 | 67 | 74 | 83 | 77 |

(grass, sedge and forb) to represent structural traits that can influence restoration of wetland (*Taft & Kron, 2014*). Statistical analyses were performed in R software. The diversity indices and DCA were analysed in the R package 'vegan' (*Oksanen et al., 2017*). One-way analysis of variance (ANOVA) was used to compare the different species diversity index. The Tukey-Kramer procedure followed the ANOVA for the post hoc comparison at $\alpha = 0.05$.

## RESULTS

### Species composition and chorological spectra

Across all study sites, we recorded a total of 103 species in 68 genera in 34 families (Appendix S1). There were 69 wetland species and 34 non-wetland species (Appendix S1). All numbers decreased gradually with time since abandonment (Table 1).

At sites abandoned <5 years prior, we recorded 62 species. Dominant species were mainly the lowland paddy weeds *Echinochloa crus-galli*, *Bidens tripartita* and *Arthraxon hispidus*.

At sites abandoned 5–15 years prior, we recorded 41 species. Dominant species included the annual species *Polygonum thunbergii*, and *Murdannia keisak* and the perennial species *Glyceria spiculosa* and *Scirpus orientalis*.

At sites abandoned >15 years prior, we recorded 34 species. Dominant species included perennial wetland species such as *C. rostrata*, *C. vesicaria*, *Phragmites australis* and *G. spiculosa*. The species composition was consistent with that in natural wetlands.

In the natural wetlands we recorded 37 species. Dominant species included the perennial species *C. rostrata*, *S. orientalis*, *G. spiculosa* and *Zizania latifolia* and the annual species *Salvinia natans* and *P. thunbergii*. The chorological spectrum is the geographical distribution of plant species. The chorological spectrum composition of plant species in our study was analyzed from the proportional distribution of chorological types in the spectrum (Fig. 1 and Table S1). The most common chorological type in the lower Tumen River Basin was Palaearctic, which accounted for 40.3% to 48.8% of species among sites (Fig. 1 and Table S1). The next most common type was Holarctic, with 22.6% to 29.2% of species. Other types were only rarely present and their proportion didn't exceed 20%.

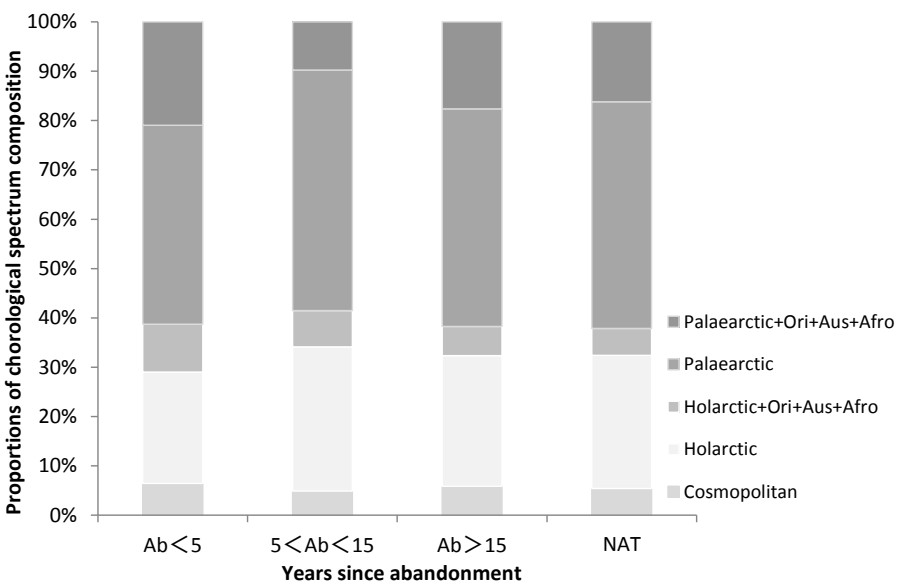

**Figure 1** **Proportions of chorological spectrum composition in paddy fields of different ages since abandonment (Ab, years) and in natural wetlands (NAT) in the lower Tumen River, northeast China.** Abbreviations: Ori, Oriental, Aus, Australian, Afro, Afrotropical.

## Species richness, diversity and evenness

Species richness and diversity were highest at <5 years since abandonment and declined with time, and were lowest in natural wetlands (Fig. 2 and Table S2). Margalef's R was significantly greater at <5 years than later and than in natural wetlands ($P < 0.001$) (Fig. 2A and Table S2). Shannon's H and Simpson's D were both non-significantly greater at <5 years than in natural wetlands (Figs. 2B, 2C and Table S2). However, Pielou's J did not change with succession (Fig. 2D and Table S2).

## SDR analysis and DCA ordination

SDR-simplex analyses showed that the point patterns are concentrated on the top side of the triangles in the different abandoned paddy fields (Fig. 3 and Table S3). Paddy fields at different times since abandonment had a high abundance replacement (R), and low Ružička similarity (S) and abundance richness difference (D). This suggests that plant communities in lowland marsh-type wetland, abundance replacement was the main component (R(A) = 83.01%; R(B) = 79.86%; R(C) = 77.06%) with relatively high values of $\beta$-diversity indicated by a low proportion of Ružička similarity (S(A) = 6.01%; S(B) = 6.98%; S(C) = 11.54%). In addition, contributions of abundance richness difference (D(A) = 10.98%; D(B) = 13.17%; D(C) = 11.40%) were similar to that of compositional similarity. Accordingly, in paddy fields at different times since abandonment had the high richness agreement (S + R) and β-diversity (R + D) and a low nestedness (S + D) (Fig. 3 and Table S3).

In the DCA ordination based on the vegetation coverage data, Eigenvalues of axes 1 and 2 were 0.8689 and 0.7182, respectively (Fig. 4 and Table S4). Natural wetlands and older

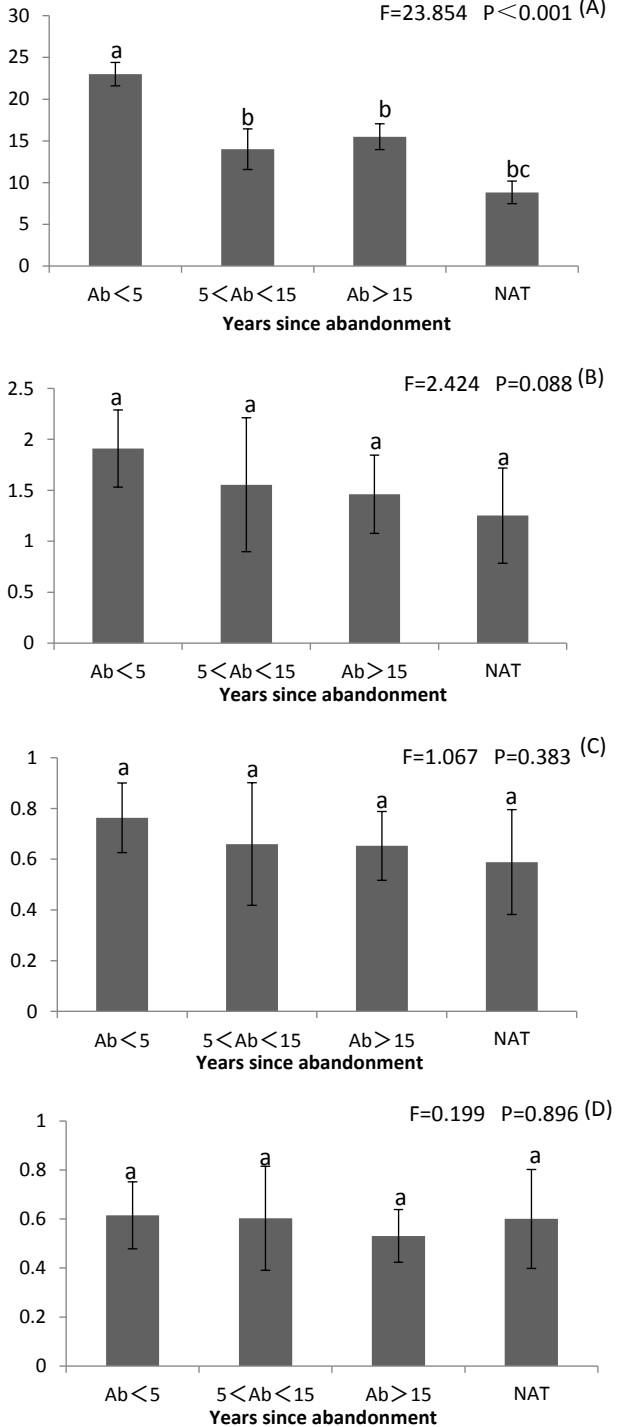

**Figure 2  Richness index (R), Shannon–Wiener diversity index (*H*), Simpson's index (D) and Pilou's evenness index (J) of plant communities in paddy fields at different times since abandonment and in natural wetlands.** The vertical bar is SD and the letters represent significant differences from post hoc Tukey tests ($P < 0.05$). (A) Richness index R; (B) Shannon–Wiener diversity index *H*; (C) Simpson's index D; (D) Pilou's evenness index J.

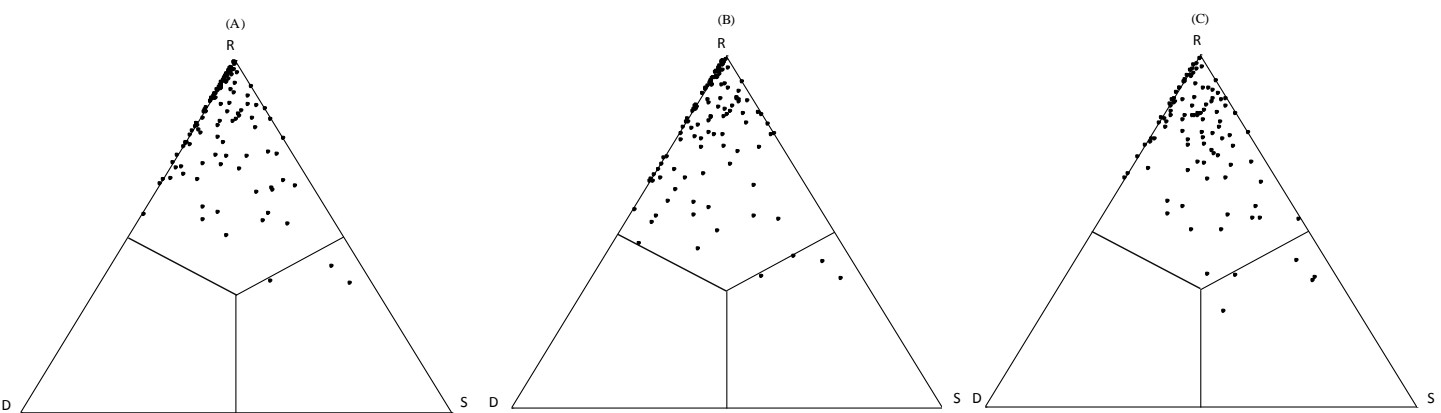

**Figure 3** SDR abunSimplex ternary plots for the cover date of different abandoned paddy fields. S: Ružička similarity; R: abundance replacement; D: abundance richness difference. (A) −5 year abandoned paddy fields; (B) 5–15 year abandoned paddy fields; (C) 15-year abandoned paddy fields.

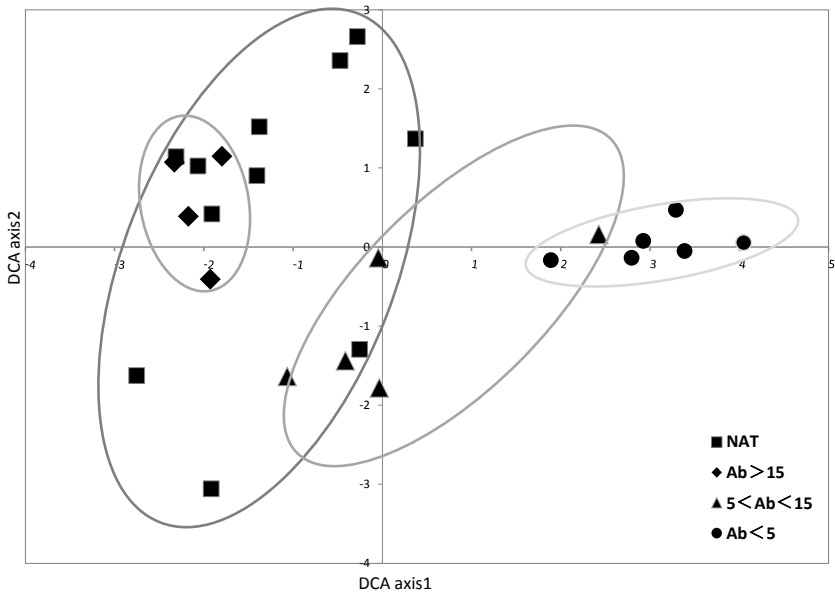

**Figure 4** DCA ordination of species percentage cover showing samples. Axes 1 and 2 accounted for 0.8687 and 0.7182, respectively, of total variation in data. Ellipses were drawn to group the samples by abandoned ages in paddy fields and reference natural wetland.

abandoned rice fields were located to the left of axis 1, and younger ones to the right. Sites tended to be clustered in relation to time since abandonment.

## Characteristics of plant functional groups

While herbs were ubiquitous, we recorded no shrubs or woody species. The vegetation community was dominated by a ground layer of forbs, sedges and grasses. The proportion of forb species in abandoned paddy fields and natural wetlands was higher than other

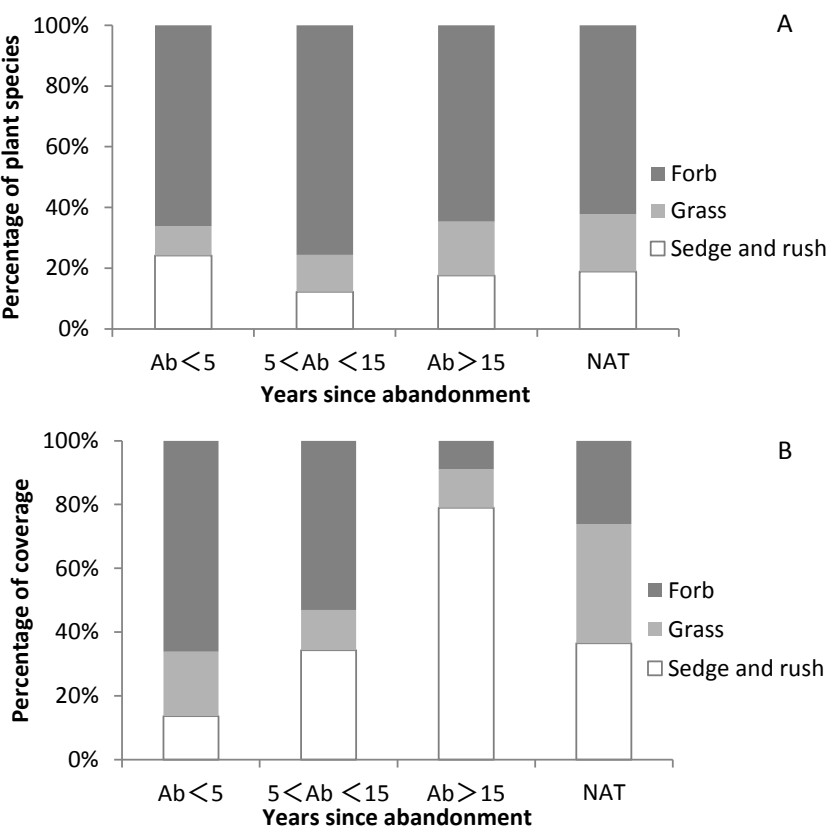

**Figure 5** Proportions of species and coverage of three functional groups in paddy fields at different times since abandonment and in natural wetlands. (A) Percentage of plant species of three functional groups; (B) percentage of plant coverage of three functional group.

plant functional groups (Fig. 5A and Table S5).The proportion of grass species in natural wetland was higher than that in abandoned paddy fields and increased with time since abandonment. The proportion of sedge and rush species was highest at <5 years since abandonment, lowest at 5–15 years, and then gradually increased again. The proportion of sedge and rush coverage was lowest at <5 years since abandonment, and then increased with the successional time (Fig. 5B and Table S5). The proportion of grass coverage in natural wetland (37.4%) was higher than in abandoned paddy fields, and with successional time it generally decreased. The proportion of grass coverage in the abandoned paddy fields was 20.2%, 12.5% and 12.1% respectively. The proportion of forb coverage was highest at <5 years since abandonment, and declined with time. After 5 years of abandonment, the proportion of sedge and grass coverage surpassed 50%, becoming the dominant plant functional groups.

Figure 6 showed that the occurrence rate of wetland species increased with the time since abandonment, and was higher in natural wetland than in abandoned paddy fields (Fig. 6 and Table S6).
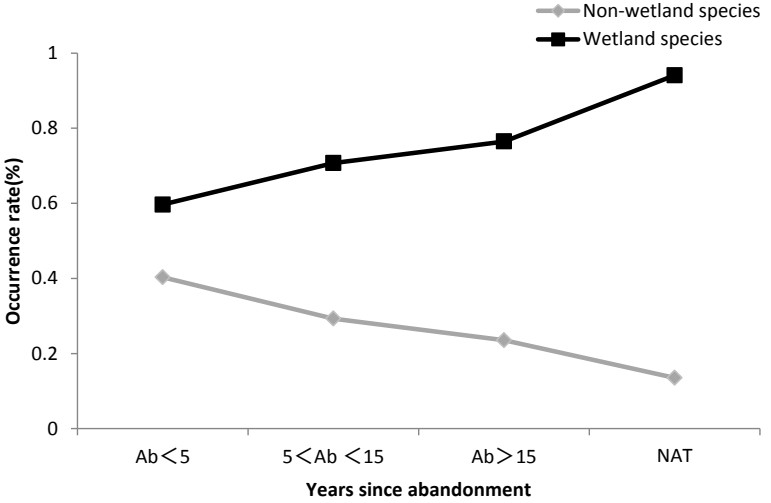

**Figure 6 Occurrence rate of wetland species and non-wetland species.** The black line indicates non-wetland species, and the gray line indicates wetland species.

## DISCUSSION

Following the abandonment of paddy fields, the vegetation undergoes a period of self-reorganization during which the fields convert to marsh-type wetland by natural succession as the species composition and dominance change over time.

We focused on abandoned paddy fields originally converted from marshland dominated by herbaceous plants in a temperate region of China. At <5 years since abandonment, paddy weeds such as the annual *E. crus-galli* and *B. tripartita* grew rapidly and became pioneer species. The dominant tussock sedge *C. rostrata* had not yet appeared, probably owing to insufficient water availability and time. At 5–15 years, the annual wetland species *Polygonum persicaria* and *M. keisak* were dominant. At >15 years, however, the habitat had become more suitable for the development of wetland communities, and the community composition gradually evolved. The plant community was strongly dominated by *C. rostrata* and *Scirpus orientalis*, unlike at <5 years. Plant richness declined but the proportion of perennial plants increased, and the dominant species were very similar to those in natural wetlands, likely because our study sites were near natural wetlands.

A clear successional trajectory appeared whereby the assemblage dominance shifted from the annual grass weed *E. crus-galli* to the annual wetland species *P. persicaria* to the tussock sedge species *C. rostrata*. These results are consistent with the theories of self-design and of secondary succession: after a wetland is destroyed, the plant community could recover naturally through an obvious process of vegetation succession (*Wang et al., 2012a*; *Wang et al., 2012b*). We therefore conclude that the wetland vegetation at our study site was adaptable to the environment and could form a stable community after 15 years of natural recovery of abandoned paddy fields.

Plant composition clearly changes with age since abandonment (*Cramer & Hobbs, 2007*). It is usually possible to distinguish three stages of succession in abandoned paddy

fields: (1) the early stage is marked by the dominance of herbs; (2) the middle stage is marked by herbs and shrubs; and (3) the late stage is terminated by woodland plants (*Lee & Kim, 1995*). In our study region on a lowland plain, however, abandoned paddy fields were marked by various herbaceous communities, including tussock meadow, as climax communities. In this type of landscape, soil water levels tend to remain high, perhaps inhibiting establishment of woodland or forest (*Shimoda & Suzuki, 1981*; *Yabe & Numata, 1984*; *Shimoda, 1987*; *Shimoda, 1996*). Previous studies of abandoned paddy fields in Japan (*Shimoda & Suzuki, 1981*; *Shimoda, 1987*; *Shimoda, 1996*) and Korea (*Kim & Nam, 1998*; *Lee, You & Robinson, 2002*) support a hydrosere model of succession. As the level of soil is progressively raised above the water level by the accumulation of humus and soil particles, the habitat becomes drier and vegetation passes from grassland to shrub and forest stages. Whether old fields further develop into shrubland or woodland depends on water conditions and surrounding vegetation (*Dovčiak, Frelich & Reich, 2005*).

On the temperate lowland plain of the lower Tumen River, the Palaearctic and Holarctic chorological types made the highest contributions. Although species composition changed with the progression of secondary succession, the chorological composition did not change conspicuously between species probably owing to the similar geological history and ecological environment. We will consider the chorological spectra in large-scale areas in future research.

## Changes in plant diversity and evenness during secondary succession

Our results showed a steady decrease in species richness and diversity with time. Both were initially high because of the cultivation of many annual plants on drained wetlands. Some surviving annual paddy weeds initially became dominant, outcompeting other species (*Yamada et al., 2007*), and some non-aquatic species were gradually eliminated owing to their inability to compete in the aquatic environment. These results are consistent with other studies: *Guo et al. (2017)* reported that plant diversity declined with time since abandonment; and *Zhang & Dong (2010)* and *Wang et al. (2017)* showed that community succession hastened initially and then slowed. Other studies in marshland and coastal wetland ecosystems show that plant communities dominated by few species tend to be more stable (*Wang et al., 2012a*; *Wang, Middleton & Jiang, 2013*). The analysis of $\beta$-diversity components revealed that species cover replacement seems to be the main process ruling community assembly. Partitioning $\beta$-diversity was an informative approach to clarify the influence of land-use histories on plant communities. In community structure under natural secondary succession, species tend to replace each other most probably due to the sites conditions together with environmental factors.

Successional age affected the recovery and development of the vegetation. The first DCA axis was closely related to the time since abandonment, which is important for natural restoration of vegetation and ecosystems (*Heshmatti & Squires, 1997*; *Martinez, Vazquez & Sanchez, 2001*). Thus, successional age was a key factor in controlling species composition and diversity (*Zhang & Dong, 2010*). We conclude that the wetland vegetation in our study region is on a positive succession trajectory, and predict that the plant communities

will eventually transition to stable communities in time. In the current study, natural wetland dominated by *C. rostrata*, *S. orientalis* and *G. spiculosa* as a reference selected in lower Tumen River Basin had its regional characteristics, such as low species diversity. Even though species diversity is low, it can still sustain itself without any intervention. The results showed that natural restoration could restore the dominant wetland species (*C. rostrata*, *S. orientalis*) like that in natural wetland. If species composition and other structural elements can be predicted, restoration practice projects could be focused on the conservation of key species (*Walker & Moral, 2003*). Therefore, in this study area, during the wetland restoration, increasing these species could be considered as a positive indicator. If species composition and other structural elements can be predicted, restoration projects can focus on key species (*Walker & Moral, 2003*).

## Changes in plant functional trait

Natural succession on abandoned paddy field could restore the dominant communities like in natural wetland, but the goal of a full recovery of an ecosystem to a pre-disturbance state is often unrealistic (*Walker, Walker & Hobbs, 2007*). Our results documented that the time required for the recovery of abandoned paddy fields to be similar in composition and cover to natural wetland will greatly exceed 15 years. However, the proportion of species and coverage of plant functional groups was different between the abandoned paddy field and natural wetland. Early stages of plant community development in abandoned fields included a high coexistence of different plant functional types, which resulted in high species richness. Perennial sedges and grasses may have competitively suppressed other functional types, which resulted in low species richness in late abandoned paddy fields. Although forb species comprised most species richness in abandoned paddy fields, they could decline with increased dominance of perennial grass and sedge species. In the late successional stage, plant communities were dominated by one to three sedge and grass species which accounted for 80% of the total plant coverage. Sedges and grass, as dominant vegetation groups in temperate marsh ecosystems, played a vital role in succession. In our study, we found that grass and sedge species rather than forbs were identified as key factors affecting succession of abandoned paddy fields on lower of Tumen River Basin. Therefore, monitoring progress toward restoration goals has tended to focus on the response of main plant functional groups. The changes in plant functional group could be used as a proxy to investigate the links between wetland species and restoration time on regional scales (*Duckworth, Kent & Ramsay, 2000*; *Voigt, Perner & Jones, 2007*). Special attention must be paid to those functional groups that showed differences in species richness and species coverage under natural succession, as these could be useful indicators of land-use history for managers of natural areas.

## CONCLUSIONS

The vegetation ecological characteristics changed during secondary succession of marshland vegetation on the lower Tumen River. Communities initially dominated by annual weeds became dominated by *Polygonum thunbergii* and then *Carex* spp. with succession. The vegetation of the abandoned paddy fields recovered through natural succession. With the

successional time, especially 15 years since abandonment, the species composition and diversity was becoming similar to the natural wetland; however, there were differences in plant functional groups. Although forb species compose most species richness in abandoned paddy fields, they can decline with increased dominance of perennial grass and sedge species. Although sedge and grass species compose lower species richness than other functional groups, they accounted for the majority of coverage. Our study was limited by the rough division of secondary succession phases due to our lack of information on the exact age of abandoned paddy fields. Therefore, we suggest that flexible time interval should be considered in future research. Future research should explore the ecological vegetation characteristics of marshlands along with natural succession driven by environmental perturbation.

# ACKNOWLEDGEMENTS

We would like to thank Haicheng Zhou for his help with plant identification and fieldwork.

## Funding
This work was supported by the National Natural Science Foundation of China (Grant No. 41771109). The funders had no role in study design, data collection and analysis, decision to publish, or preparation of the manuscript.

## Grant Disclosures
The following grant information was disclosed by the authors:
National Natural Science Foundation of China: 41771109.

## Competing Interests
The authors declare there are no competing interests.

## Author Contributions
- Guanglan Cao conceived and designed the experiments, performed the experiments, analyzed the data, contributed reagents/materials/analysis tools, prepared figures and/or tables, authored or reviewed drafts of the paper, approved the final draft.
- Kazuaki Tsuchiya conceived and designed the experiments, authored or reviewed drafts of the paper.
- Weihong Zhu contributed reagents/materials/analysis tools, authored or reviewed drafts of the paper.
- Toshiya Okuro conceived and designed the experiments, contributed reagents/materials/analysis tools, authored or reviewed drafts of the paper, approved the final draft.

## Field Study Permissions
The following information was supplied relating to field study approvals (i.e., approving body and any reference numbers):

Field experiments were approved by the landowners and village head. The landowner's name is Jin Xuemin and the village head's name is Hu Naifeng.

## Data Availability

The raw data are available in a Supplemental File.

## Supplemental Information

Supplemental information for this article can be found online at http://dx.doi.org/10.7717/peerj.6704#supplemental-information.

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
