# Peer review of "Vegetation dynamics of abandoned paddy fields and surrounding wetlands in the lower Tumen River Basin, Northeast China"

_PeerJ, doi:10.7717/peerj.6704_

## Round 0.1 · original submission · Major Revisions

Although reviewers found that this paper was meritorious, they have made some criticisms of your research, and suggest major revisions to your manuscript. The general assessment of the reviewers is that further statistical analyses are required to support the conclusions. Please in figure 2 delete the text near the objects and clarify the meaning of the grey lines.

·

Basic reporting

On my opinion this MS is written in a clear and fluent English. The style is concise and scientifically sound. Literature references are sufficinetly updated in spite I will suggest to add other methods for the measurement of biodiversity. Tables and figureas are clear and appropriated. I am not expert in E Asian floras so I do not know if all the names used are updated and currenlty accepted.

Experimental design

The main gap of this Ms is that the authors have chosen to measure biodiversity using indices that take into account only the number of species without taking into account the "biogeographic value" of these species. In this way it comes out the strangeness that immediately after the abandonment of the cultivation there is a greater biodiversity compared to the plots abandoned for a longer time. In reality, by experience, a decrease in the total number is likely to occur, but an increase in species with higher biogeographical value (with a smaller distribution) Therefore I would like to add to the observations made also the chorology of the species found and add the analysis of the chorological spectra to those already performed.

Validity of the findings

The addition of the analysis suggested, in my opinion will make more robust the conclusions on the data collected.

Additional comments

I suggest to accept the Ms after the inclusion of a chorological analysis of the species recorded.

·

Basic reporting

The manuscript is clearly written even if some grammar errors (or typos) are present, for examples: lack of s in the plurals, concordance of the times not always adequate, mash instead of marsh, lack of space between words, C. rostrate instead of C. rostrata, etc. (but I am not a mother tongue, so I am not able to judge English language style).
References must be accurately checked: for example, Duckworth et al., 2000 is quoted in the text but not in the reference list; the same for Luisa et al. 2001; Voigt et al. is not in alphabetical order in the reference list, etc.
The study poses interesting questions, but there are also some basic weaknesses. The main one concerns the statistical treatment of data, which is rather simplistic and not thorough.
In general: all the analyses rely on an important data set which is not shown or available in detail. Appendix A is informative but not sufficient. I think that it is crucial to add the consultable dataset as supplementary material.

Experimental design

Survey design
secondary succession phases of <5 years, 5–15 years and >15 years since the last cultivation: acceptable and, at present, uneditable, but this subdivision could involve some problems: for example, an analysis of a first phase of 4 years compared with a second phase of 5 years could be misleading. If you have data, a better division between phases could be <5 years, 8–12 years and >15 years

Data analysis
The indices used are acceptable, sometimes even redundant, but they are not sufficient to explore species-richness and species-diversity patterns during a succession. I suggest to analyze also the beta diversity, take a look to the following references:
Podani, J., & Schmera, D. (2011). A new conceptual and methodological framework for exploring and explaining pattern in presence–absence data. Oikos, 120(11), 1625-1638.
Podani, J., Ricotta, C., & Schmera, D. (2013). A general framework for analyzing beta diversity, nestedness and related community-level phenomena based on abundance data. Ecological Complexity, 15, 52-61.
Baselga, A., Orme, D., Villeger, S., De Bortoli, J., & Leprieur, F. (2017). Partitioning beta diversity into turnover and nestedness components. Package betapart, Version, 1-4.
Baselga, A., & Leprieur, F. (2015). Comparing methods to separate components of beta diversity. Methods in Ecology and Evolution, 6(9), 1069-1079.

Validity of the findings

Results
Must be reformulated and implemented after the changes in M&M procedures.
In the present text: “With the successional time, the occurrence rate of wetland species generally increased”, and in Fig. 4: state what are the wetland species and not-wetland species, and why, this is not evident in the data set.

Discussion and Conclusions
Must be reformulated and implemented after the changes in M&M procedures.
In the present text, lines 219-226: discuss better why a marsh woodland is not the climax or Potential Natural Vegetation of the study area.

---

## Round 0.2 · Minor Revisions

The reviewers found the text strongly improved. Nevertheless, a reviewer suggests you to take into account not only similarity but all component of beta diversity, so species replacement/turnover and richness difference. In particular, he suggests you to use the SDR approach implemented by Podani. I also recommend you to delete the text near the objects in figure 2 (the meaning of symbols is in the legend) and to clarify the meaning of the grey lines.

Once again, thank you for submitting your manuscript to PeerJ and we look forward to receiving your revision.

Sincerely,
Gabriele Casazza

·

Basic reporting

I have seen that the implementations requested were applied by the authors.

Experimental design

Now the experimental design is more convincing than in the previous submission.

Validity of the findings

The authors bettern demonstrated the validity of their findings.

Additional comments

All the requested implementations were applied.

·

Basic reporting

The manuscript has been improved and almost all the changes suggested have been accepted.
Nonetheless, the basic weakness concerning the statistical treatment of data is not completely solved.

Experimental design

The authors use only Jaccard’s similarity index to describe beta diversity, but as showed by several studies (i.e. those referenced in my first review) similarity (S) is only one of the beta diversity components, together with species replacement/turnover (R) and richness difference (D). A SDR approach, or something similar, would be more appropriate to investigate and highlight what really happens during succession.
I think that authors must implement this aspect to improve definitively the manuscript.

Validity of the findings

The present findings are acceptable, but if implemented with the analysis above suggested, results and comments will be more sound and interesting

---

## Round 0.3 · accepted · Accept

All changes that was solicited from the reviewer were done. So, I am very pleased to say that your paper "Vegetation dynamics of abandoned paddy fields and surrounding wetlands in the lower Tumen River Basin, Northeast China " is accepted for publication in the PeerJ

#